# Thermodynamic-driven polychromatic quantum dot patterning for light-emitting diodes beyond eye-limiting resolution

Tae Won Nam[1], Moohyun Kim[1], Yanming Wang [2], Geon Yeong Kim [1], Wonseok Choi[1], Hunhee Lim[1], Kyeong Min Song[1], Min-Jae Choi[1], Duk Young Jeon[1], Jeffrey C. Grossman [2] & Yeon Sik Jung [1✉]

The next-generation wearable near-eye displays inevitably require extremely high pixel density due to significant decrease in the viewing distance. For such denser and smaller pixel arrays, the emissive material must exhibit wider colour gamut so that each of the vast pixels maintains the colour accuracy. Electroluminescent quantum dot light-emitting diodes are promising candidates for such application owing to their highly saturated colour gamuts and other excellent optoelectronic properties. However, previously reported quantum dot patterning technologies have limitations in demonstrating full-colour pixel arrays with sub-micron feature size, high fidelity, and high post-patterning device performance. Here, we show thermodynamic-driven immersion transfer-printing, which enables patterning and printing of quantum dot arrays in omni-resolution scale; quantum dot arrays from single-particle resolution to the entire film can be fabricated on diverse surfaces. Red-green-blue quantum dot arrays with unprecedented resolutions up to 368 pixels per degree is demonstrated.

[1] Department of Materials and Science and Engineering, Korea Advanced Institute of Science and Technology, 291 Daehak-ro, Yuseong-gu, Daejeon 305-701, Republic of Korea. [2] Department of Materials Science and Engineering, Massachusetts Institute of Technology, Cambridge, MA 02139, USA. ✉email: ysjung@kaist.ac.kr

Among various light-emitting materials for next generation displays, quantum dots (QDs) are prominent due to their unique optoelectronic properties, such as their high brightness and narrow emission spectra, extensive colour tunability, high quantum yield, and good stability[1–17]. Extensive research has been conducted to realise practical electroluminescent active matrix QD light-emitting diodes (ELQLEDs)[1,7–9,11,18]; although the device efficiency of individual red-green-blue (RGB) ELQLEDs have been sharply increasing for the past decades, patterning RGB full-colour QD arrays still remains as a critical bottleneck. The complexity of QD patterning is mainly due to the colloidal state of pre-synthesised QDs. Solution-state QDs are not compatible with well-established vacuum deposition processes such as thermal evaporation, which are widely used in the manufacturing of organic light-emitting diodes (OLEDs). Several alternative fabrication techniques, such as transfer-printing, inkjet printing and lithography[17,19–30] have been utilised to demonstrate the patterning of colloidal QDs with the goal of improving the performance of conventional displays.

The prospects of ELQLEDS can further extend to the next-generation "near-eye" devices such as head-mounted displays and smart glasses for virtual reality (VR) and augmented reality (AR) applications, which inevitably require a significant leap in the number of pixels per degree (PPD) due to the much shorter viewing distance[31]. The increase in pixel density must be followed by a replacement of emissive material with wider colour gamut to maintain accurate colour expression[32]. QDs are promising for such upcoming platforms with ultrahigh resolutions due to their highly saturated colour gamut and the wearable and transparent display applicability of thin QD films. Considering that the sub-pixel feature size in current smartphones is in the range of tens of micrometers (10–30 PPD depending on the viewing distance when applied for VR), a feature size of >150 PPD displays would be required to scale down to the sub-micrometer level. However, previous QD patterning technologies have limitations in achieving uniform full-colour pixel arrays with such ultrahigh resolutions and high fidelity levels. More importantly, the light-emission properties of QDs are often degraded to the degree of insignificance after their patterning, which is a critical drawback.

Here, we present an immersion transfer-printing (iTP) technique, which allows solution-based deposition, patterning, and printing of QD arrays in omni-resolution scale; QD arrays with even single-QD resolutions to the entire film can be fabricated and printed onto diverse substrates. Using iTP, an individual RGB pixel array can be aligned and printed to realise a full-colour QD array with a pixel density of >350 PPD, which is the highest QD pattern resolution reported to date. As opposed to the kinetic-driven mechanism of conventional elastomer-based contact printing, programmed QD self-assembly on hard templates combined with a thermodynamic-driven binary adhesion switching mechanism is the key principle of iTP, which is supported by molecular dynamics (MD) simulation results. Utilisation of the technique provides unprecedentedly high resolutions and near 100% printing yields while also preserving the optical quality of the QDs, which is supported by outstanding current efficiency and the external quantum efficiency (EQE) of ELQLEDs fabricated using iTP.

Moreover, as the printing cycle is repeated, gradual degradation of the stamp topography such as sagging and leaning restricts reproducibility in producing defect-free pixel arrays[23].

In order to realise a universal fabrication methodology for well-defined QD arrays throughout the entire resolution range, we developed iTP as a unique patterning technique that can integrate deposition, patterning, and printing of colloidal QDs. Figure 1a illustrates the overall fabrication procedure. A hard master template is prepared using KrF photolithography and plasma etching. The surface of the Si template is then modified by a short-chain hydroxyl-terminated PDMS (3–4 nm thickness by ellipsometry) to minimise its surface energy for reliable detachment of the QDs from the template. Solutions containing various QDs capped with oleic acid are spun-cast and simultaneously patterned on the template via programmed wetting/dewetting mechanism.

To securely detach the patterned QD film from the Si template, poly(methyl methacrylate) (PMMA) is spun-cast on the QD-patterned substrate as a sacrificial transfer medium. Polyimide (PI) adhesive film is subsequently laminated on the PMMA layer followed by peeling of the patterned QD film away from the master. The uniformly picked QD pattern array is then brought into a contact with a target substrate with a minimal contact pressure of <2 kPa and the final transfer phenomenon is carried out during the immersion in acetone until the PMMA/PI is completely removed from the substrate. From the above processes, the one-step programmed self-assembly and the immersion printing (dotted magenta boxes in Fig. 1a) are the two key stages enabling the effective patterning of QDs.

Depending on the resolution of the master template, iTP successfully produced diverse QD pattern arrays in a wide range of resolution from even single-QD-width pattern to totally continuous QD films, as shown in Fig. 1b. In particular, to achieve single-QD-particle resolution of printed QD array (Fig. 1b, left), a sub-10 nm resolution hard template was prepared using directed self-assembly of polystyrene-*b*-polydimethylsiloxane block copolymers[30,33–37] (Supplementary Fig. 1). Figure 1c, d and Supplementary Fig. 2 demonstrate application-specific design of Si template, and patterned and printed QD arrays. A master template with a 60 nm depth trench was used to pattern and print a QD film pattern with 30–40 nm thickness (Fig. 1h). As reported in previous studies, the fidelity of contact-printed QD patterns using elastomeric molds diminishes during repeated transfer processes, and exacerbates in a higher resolution regime[23]; propagation of cracks induced by applied stress during retrieval and stamping steps causes pattern discrepancy near the edges. Such exertion of pressure as in the case of conventional contact printing is absent or minimal in the steps of iTP, which enables ~100% structural and topological fidelity while achieving a small pattern edge roughness of <10 nm, as supported by the topographic AFM images and profile characterization results in Fig. 1e–h. The outstanding uniformity of the printed QD film thickness of the micrometer-scale pixels is also demonstrated in Supplementary Fig. 3. Although we selected 40 nm QD film thickness appropriate for the ELQLED application, the applicability of the technique for higher film thickness is also demonstrated using deeper trench (250 nm) to pattern 150 nm thick QD films as shown in Supplementary Fig. 4.

## Results

**Overall procedure of omni-resolution iTP.** For conformal contact and printing on a large-scale area, conventional contact-printing techniques utilise soft elastomeric molds (e.g. polydimethylsiloxane (PDMS)) on which topographic pixel information is written. However, the usage of a PDMS mold typically limits replication resolution and printing fidelity[17,18,20,21,25].

**QD patterning via programmed wetting and dewetting.** Patterning colloidal QDs is challenging because a top-down method based on vacuum deposition is not applicable. Instead, of few alternative methods, in contact printing, a PDMS mold with protruding patterns is put into contact with an unpatterned QD film to separate QD patterns. However, the adhesion strength of

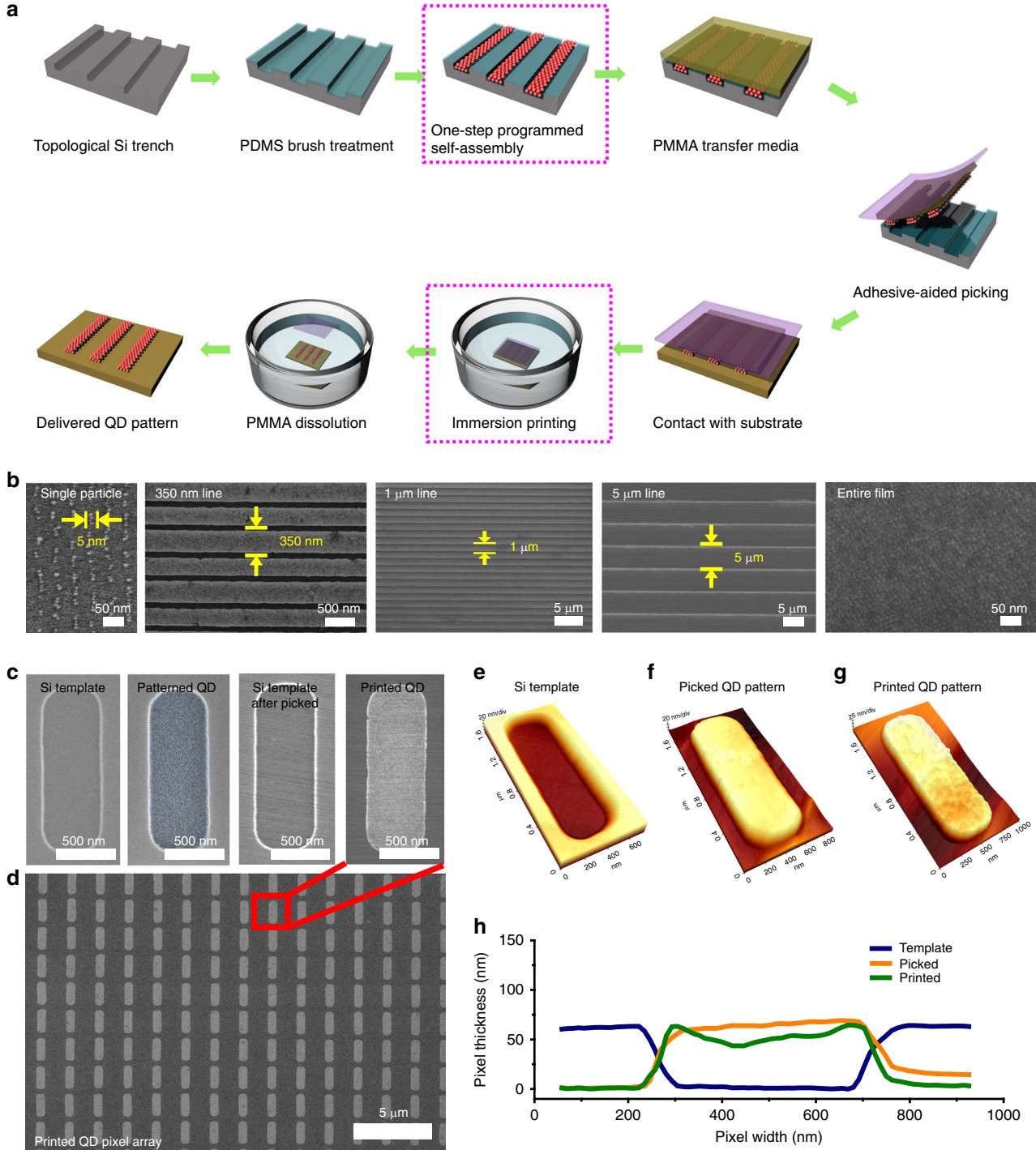

**Fig. 1 Patterning and printing of QD films using iTP. a** Schematic of iTP process. The two key stages of iTP are emphasized with dotted magenta boxes. **b** SEM images of printed QD arrays in different resolutions: single-particle, 350 nm, 1 μm, 5 μm, and entire film (from left to right). **c** SEM images of topographic trench template fabricated on Si substrate, patterned QD film in trench domain, the Si template after QDs are picked, and printed QD pattern (from left to right). **d** SEM image of printed QD pixel array (500 nm pixel width). Topographic AFM images of the Si template (**e**), picked (**f**), and printed (**g**) QD pixel. **h** AFM thickness profile plot of the Si template, picked and printed pixel.

QD/PDMS is not sufficiently high, and thus the delamination of QDs from the substrate is achieved by a rapid "kinetic" detachment of the stamp, which seriously limits the patterning yield, controllability, and fidelity[20]. In order to resolve these challenges, we introduce a domain-specific, in situ assembly of QDs in a "hard" template, which is a paradigm-shifting bottom-up approach, enabling clear-cut QD patterns along the well-defined trench walls during the deposition stage. The most critical

challenge here is selective wetting of the QD solution in designated trench domains and complete removal of QDs on the mesa area by inducing dewetting, which are mutually exclusive behaviors of a solution spun-cast on a surface.

Such a programmed wetting and dewetting phenomenon is realised based on precise control of solution wettability and surface topography on the substrate. Heptane was selected as the majority solvent in the QD solution because it enables both

excellent dispersion of oleic-acid capped QDs and simultaneously a good wetting property on the PDMS-functionalised surface of the hard template. However, a pure heptane QD solution provides only uniform QD films on the entire surface of topographic templates without area selectivity. Thus, we systematically engineered the wetting behavior of the QD solution on the PDMS-treated Si template by introducing an additional solvent in the solution, which provides a relatively higher tendency towards dewetting on the surface.

To determine constituents of the binary solvent, we theoretically estimated the stability of a liquid thin film on a solid surface using a previously reported theoretical model of the effective interface potential[38]. Assuming that our Si/SiOx/PDMS/solvent/ air system is a multi-layered dielectric, the effective interface potential composed of van der Waals interactions may be expressed as follows:

$$\Phi(h)_{vdW} = -\frac{A}{12\pi h^2} \qquad (1)$$

where $A$ is the Hamaker constant and $\Phi(h)$ is the energy per unit area as a function of the film thickness, $h$. Lifshitz theory is applied to calculate $A$ of the entire system[39]. $A$, the strength of the van der Waals forces between solid/liquid and liquid/air interfaces, for a three media system may be expressed as follows:

$$A \approx \frac{3}{4}kT\left(\frac{\epsilon_1 - \epsilon_3}{\epsilon_1 + \epsilon_3}\right)\left(\frac{\epsilon_2 - \epsilon_3}{\epsilon_2 + \epsilon_3}\right)$$
$$+ \frac{3h\nu_e}{8\sqrt{2}}\frac{(n_1^2 - n_3^2)(n_2^2 - n_3^2)}{\sqrt{(n_1^2 + n_3^2)}\sqrt{(n_2^2 + n_3^2)}\left\{\sqrt{(n_1^2 + n_3^2)} + \sqrt{(n_2^2 + n_3^2)}\right\}}$$
$$(2)$$

where $\nu_e$ is the single electronic absorption frequency and $n$ and $\epsilon$ are the refractive index and the dielectric constant of each media, respectively ($n_i^2 = \epsilon_i$, in visible spectral range, $i = 1$(air), 2(solid), 3(liquid)). Due to additivity of forces, the effective interface potential of our Si/SiOx/PDMS/solvent/air system is the sum of all the van der Waals contributions of the liquid layer with film thickness $h$ and solid layers with thickness $d_{PDMS}$ and $d_{SiOx}$. The raw data used to calculate $A$ values of each composition is provided in Supplementary Tables 1 and 2. After calculating the values of $A$ of each solid layer component, Eqs. (1) and (2) are integrated to give the approximate effective interface potential of Si/SiOx/PDMS/solvent/air system, which can be expressed as:

$$\Phi(h)_{vdW} = -\frac{A_{PDMS}}{12\pi h^2} + \frac{A_{PDMS} - A_{SiOx}}{12\pi(h + d_{PDMS})^2} + \frac{A_{SiOx} - A_{Si}}{12\pi(h + d_{SiOx} + d_{PDMS})^2}$$
$$(3)$$

The effective interface potential curve shown in Supplementary Fig. 5a suggests that heptane is stable in the Si/SiOx/PDMS/ solvent/air system, which is consistent with the superoleophilic wetting behavior (contact angle < 4º) of heptane on the PDMS-treated Si wafer surface.

Among non-polar solvent candidates that enable dispersion of oleic-acid-capped QDs, chloroform and toluene were considered as dewetting-inducing solvents because their n values deviate the most from that of PDMS, leading to $A > 0$ (instability) from Eq. (2). (selection criterion: $n_{solvent} - n_{PDMS} > 0.04$, polarity index < 4.2) This calculation indicates that both solvents exhibit a metastable film state in our system; the second derivative of $\Phi(h)$ leads to a negative value below the critical film thickness, $h_{crit}$, (evaluated using linear stability analysis), which corresponds to the spinodal dewetting phenomenon[38,40]. According to Supplementary Fig. 5b, $h_{crit}$ for toluene and chloroform was calculated to be 5.63 nm and 3.95 nm, respectively. The $h_{crit}$ values are comparable to the QD diameter (3–6 nm), which means the

solvent-tracing behavior of QDs is sensitively determined by $h_{crit}$ as the solvent evaporates, with the solution reaching the sub-10 nm thickness regime upon the last stage of spin-casting. A solvent with a high $h_{crit}$ dewets before the QDs experience the mutual lateral capillary forces needed for aggregation and immobilization, allowing the QDs to trace the dewetted morphology of the solvent. Therefore, the dewetting and removal of QDs on the PDMS-functionalised surface will be more effective with toluene, which has the higher $h_{crit}$ and will become the majority solvent component at the last stage of spin-casting (due to its lower evaporation rate than heptane). As shown in Supplementary Fig. 5c, we also confirmed consistency between the calculated and experimental characteristic wavelength, $\lambda_s$ (the periodicity of unstable modes whose amplitude grows fastest) of the spinodal dewetting phenomenon. (See Supplementary Note 1 for more details.)

For experimental verification, the heptane/toluene binary solvent QD solution is applied on topographic trench templates. The maps of coated QD morphology with varying QD concentration and solvent composition are plotted separately for trench and mesa domains in Fig. 2a, b, which reveal a boundary line between the wetting and dewetting ranges of the parameters. SEM images of representative QD film morphologies are provided in Fig. 2d–g. A complete set of SEM images showing QD film morphologies of heptane, heptane/chloroform, and heptane/toluene at different QD concentrations and solvent compositions are provided in Supplementary Fig. 6.

While unpatterned QD films are formed at an excessively high QD concentration, spinodal dewetting took place for a relatively lower QD concentration range (<20 mgml⁻¹) and higher toluene fraction. In particular, as depicted in Fig. 2a, e, as the QD concentration decreases, heterogeneous dewetting at the mesa edge regions was observed prior to the spinodal dewetting due to mass flow from the mesa to the trench at the edge[41,42]. At the center of the mesa region, however, spinodal dewetting was the predominant phenomenon. In contrast, the heptane-only solution did not exhibit dewetting behavior even at a low QD concentration as predicted by the calculation model discussed earlier.

An important observation in Fig. 2a, b is that the wetting/ dewetting boundary line for the trench is positioned at the side of the higher toluene fraction compared to that for the mesa region, which can be attributed to the topography-induced confinement effect in trenches. This strongly suggests the feasibility of programming the wetting/dewetting phenomenon at designated locations with topographic contrast via systematic control of the experimental parameters. Figure 2c shows the window of optimised toluene composition of 13.5–16.5 (shadowed in blue) at which domain coverage of QD films on the trench area is preserved at 100% but the coverage on the mesa area drops to almost 0% due to complete dewetting. A corresponding SEM image of selectively assembled QDs in the hard template is highlighted with a green box in Fig. 2g.

**Orthogonal-solvent-assisted detachment of QD patterns.** Patterned QD films can be uniformly and reproducibly released from the hard template by casting a proper sacrificial transfer-medium and solvent composition. For example, PMMA provides stronger adhesion strength at the QD/PMMA interface than that of the QD/template surface with PDMS functionalization, enabling 100%-yield detachment of QD patterns from the hard template. PMMA is selected also due to its solubility in acetone, which is an orthogonal nonsolvent to oleic-acid capped QDs. However, the use of acetone caused only partial (non-conformal) wetting behavior on the PDMS-treated Si surface with a

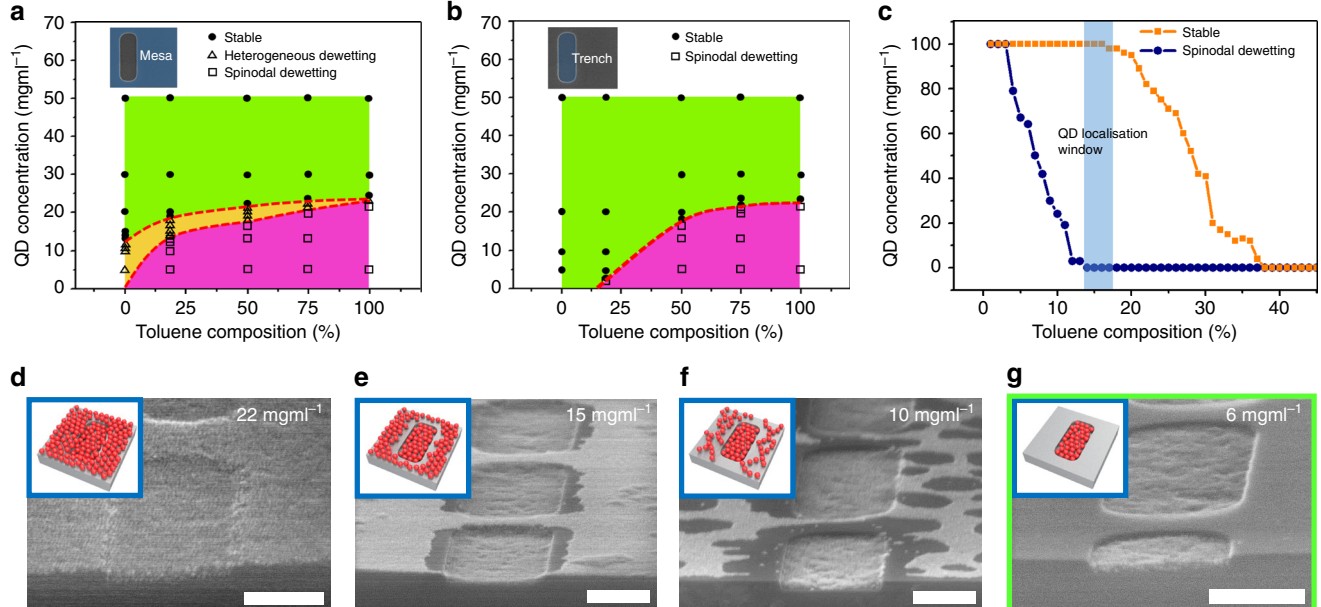

**Fig. 2 Domain specific patterning of QD films by controlled dewetting mechanism.** QD concentration and solvent composition dependent stability diagram of dried QD film are plotted separately for mesa (**a**) and trench (**b**) domains. The symbols ●, △ and □ indicate conformal QD film, heterogeneous dewetting near the mesa edge, and spinodal dewetting throughout the domains, respectively. **c** The domain coverage of QD film on mesa and trench area as a function of change in solvent composition. The blue shadow indicates experimental window of solvent composition in which the contrast in domain coverage between mesa and trench area is the greatest. **d–g** Tilted SEM images (respective schematics in the insets) that show the change in dewetting behavior of dried QD films as QD concentration decreases (constant toluene composition at 13%). Scale bars denote 300 nm.

contact angle of 19° and hence prohibits conformal coating of PMMA on a patterned QD/Si master substrate having significant hydrophobicity.

To solve this, wettability of the PMMA solution on the QD pattern substrate is engineered using an acetone/heptane binary solvent. Heptane here is the additional solvent component to facilitate conformal wetting of PMMA. Optical images of spun-cast PMMA layers coated on the QD pattern substrate at different compositions of the binary solvent are shown in Fig. 3a–c with respective schematics. As shown in the cross-section SEM image of the morphology of the spun-cast PMMA film on the QD patterned master (Supplementary Fig. 7f), a dry PMMA film with film thickness of ~140 nm forms conformal contact with the patterned QDs and the mesa surface. The film coverage of coated PMMA on the trench (QDs) and mesa (PDMS-functionalization) is plotted as a function of the heptane composition in Fig. 3d.

The transfer-printing yield and contact angle of the PMMA solution on the PDMS-treated Si surface are plotted as a function of the heptane composition, as shown in Fig. 3e. The average printing yield was calculated as the percentage of final QD film domain area divided by the initial area of the master trench domain using image analysis software (ImageJ) (average yield collected from 10 random points). The dotted lines across the two plots (Fig. 3d, e) indicate the existence of an optimised heptane composition window (12.5–16%), in which the domain coverage of PMMA on both the trench and mesa reach 100% and corresponding transfer printing yield of 100% (optical image highlighted with green box in Fig. 3b). At a heptane composition below the window, the domain coverage of PMMA film on both the QD and PDMS surface is not sufficient to provide a conformal interface over the entire substrate and the corresponding transfer printing yield also diminishes (Fig. 3a). At a heptane composition above the window, diffusion of QDs into the heptane-containing PMMA solution occurs during a comparable time period of spin-casting and would result in a significantly lower transfer yield, as shown in Fig. 3c, e.

**Thermodynamic-driven adhesion switching mechanism.** A thermodynamic-driven pressure-free transfer-printing mechanism is proposed in this study. After attaching the QD/PMMA/PI on the target receiver substrate, the sample is immersed in acetone, which is a good solvent for PMMA and a non-solvent for the oleic-acid-capped QDs. Figure 4a illustrates that the PMMA/QD interface is delaminated as the PMMA chains disentangle and diffuse in acetone and then an acetone/QD interface is newly formed. Due to negligible interaction between polar acetone molecules and nonpolar oleic-acid ligands of QDs, the adhesive energy of the interface is now significantly weakened, while that of the relatively stronger QD/target substrate is preserved.

MD simulations were conducted to support the stability of the printed QDs on the target substrate. The interaction energy between the QDs and the substrate is expected to be positively correlated with the adhesion energy[43–46], so that it could be used as a good indicator for QD surface affinity. In the MD simulation, we considered two configurations—(1) a QD contacted on Si substrate (Fig. 4b left) and (2) a separated QD from the substrate (Fig. 4b right), both of which are surrounded by acetone. The change in energy of the system before and after the QD detachment was calculated as a function of displacement of the QD as plotted in Fig. 4c. From the fitted curve, the fact that excess energy is necessary to lift the QD from the Si substrate supports the stability of the printed QD after immersion. Also, Supplementary Fig. 8c, d shows that the change in the QD displacement at the relaxed state on the Si surface surrounded by the acetone is negligible at <0.5 Å, again supporting the stability.

For conventional kinetic-driven printing, the adhesion switching point is ambiguous along continuously applied external pressure, which is due to a close competition between the two interfaces[47,48]. On the contrary, the adhesion switching in the iTP proceeds in a binary on-and-off switching mode, enabling unprecedentedly high printing yield. The binary adhesion switching mechanism enables the printing of QD patterns on substrates with much lower surface energy than that of transfer

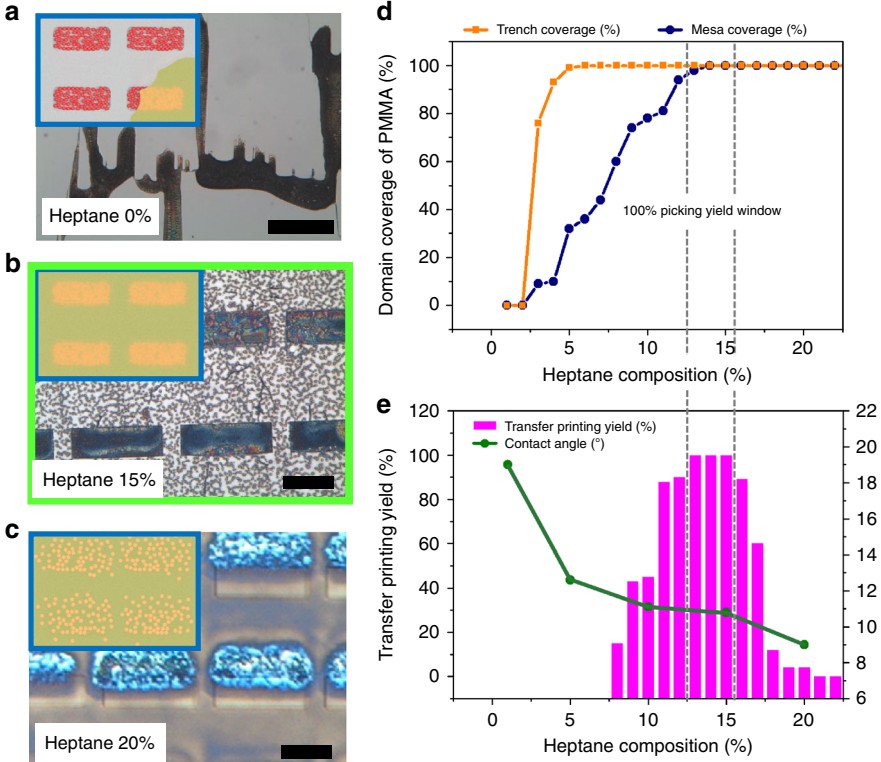

**Fig. 3 Engineering binary solvent for extracting assembled QDs using a sacrificial transfer-medium. a–c** Optical images of spun-cast PMMA film on patterned QD substrate at different heptane composition are shown. Schematics of each images are shown in respective insets. Scale bar of **a** denotes 100 μm. Scale bars of **b**, **c** denote 5 μm. **d** Domain coverages of PMMA film on trench and mesa domain, plotted separately as a function of heptane composition. **e** Transfer printing yield of patterned QDs and measured contact angles of spun-cast PMMA solution on PDMS surface are plotted as a function of heptane composition. The grey dotted lines across (**d**) and **e** indicate the optimised experimental window in which PMMA coverage reaches 100% for both trench and mesa domains and as a consequence gives 100% picking yield.

media (Fig. 4g). Figure 4d–f shows printed QD patterns on Si, PDMS, and polytetrafluoroethylene (PTFE, i.e. Teflon) surfaces, respectively. Printing yield of 74% was achieved even on the exceedingly low surface energy PTFE substrate, which is, as far as we know, the first demonstration of transfer printing on such an extremely low-energy surface.

In order to support the experimental results and explain the hypothetical relationship between competing interfaces during iTP, MD simulation was carried out to extract averaged interaction energy per area of each QD/media interface as summarized in Supplementary Fig. 8e and the calculated data are used to plot Fig. 4g. At the point of immersion, the change in the QD-PMMA interface to QD-acetone reverses the predominance of adhesion and enables effective printing on any surfaces that have higher interaction energy than that of QD-acetone. These results suggest the versatility of iTP tool in terms of application to diverse target substrates to effectively deliver desired objects. The demonstration on a wide range of target surfaces with different curvature, roughness, and hydrophobicity, is shown in Fig. 4h.

The thermodynamic-driven mechanism enables pressure-free picking and placing processes, which minimises the damage to the hard master template and hence maximises the template lifetime. As shown in Supplementary Fig. 9, repeated printing of a 1 μm width QD line pattern was carried out using conventional contact-printing and iTP. The transfer yield of conventional contact-printing significantly diminished during 100 repeating cycles, whereas the yield of iTP is still preserved for the same repeating cycles (Supplementary Fig. 9e). Moreover, the pressure-free printing mechanism enables repeated multi-layer stack printing of QD patterns without destroying the underlying

patterns, which can be applied to fabricate a large-area QD woodpile architecture, as shown in Supplementary Fig. 10. The fabrication of such a free-standing architecture composed solely of light-interacting quantum nanostructures with sub-micrometer resolution has not been demonstrated before, and we further anticipate that it will exhibit novel photonic crystal properties. In addition to the structural preservation of the iTP QD films, the change in the photoluminescence quantum yield (PLQY) of the printed films was monitored in order to investigate the effect of acetone immersion on the optical property of the QD films. The normalized PL spectra and PLQY of QD films after repeated iTP trials (Supplementary Fig. 11) show that QY of the QD film after the 3rd iTP is preserved at 93% relative to the as spun-cast QD film. A spun-cast film immersed in the same acetone environment during a single immersion cycle without PMMA transfer layer shows QY degradation down to 75%, which indicates the QD film is effectively passivated by the PMMA layer during iTP process, minimizing the damage from the solvent.

**ELQLED fabricated by iTP.** As shown in the confocal microscope images in Fig. 5a–e, polychromatic QD arrays were successfully fabricated by sequentially repeating the iTP process of individual red, green, and blue QD pixels using a mask aligner. Supplementary Movie 1 and Supplementary Fig. 12 show the demonstration of pattern alignment using a manual mask aligner and SEM images of sequentially printed pixels, respectively. Depending on the feature size of the pre-patterned trench master template, QD arrays with different resolution ranging up to 14,063 PPI were achieved, which is to the best of our knowledge the highest resolution full-colour QD array. The calculated

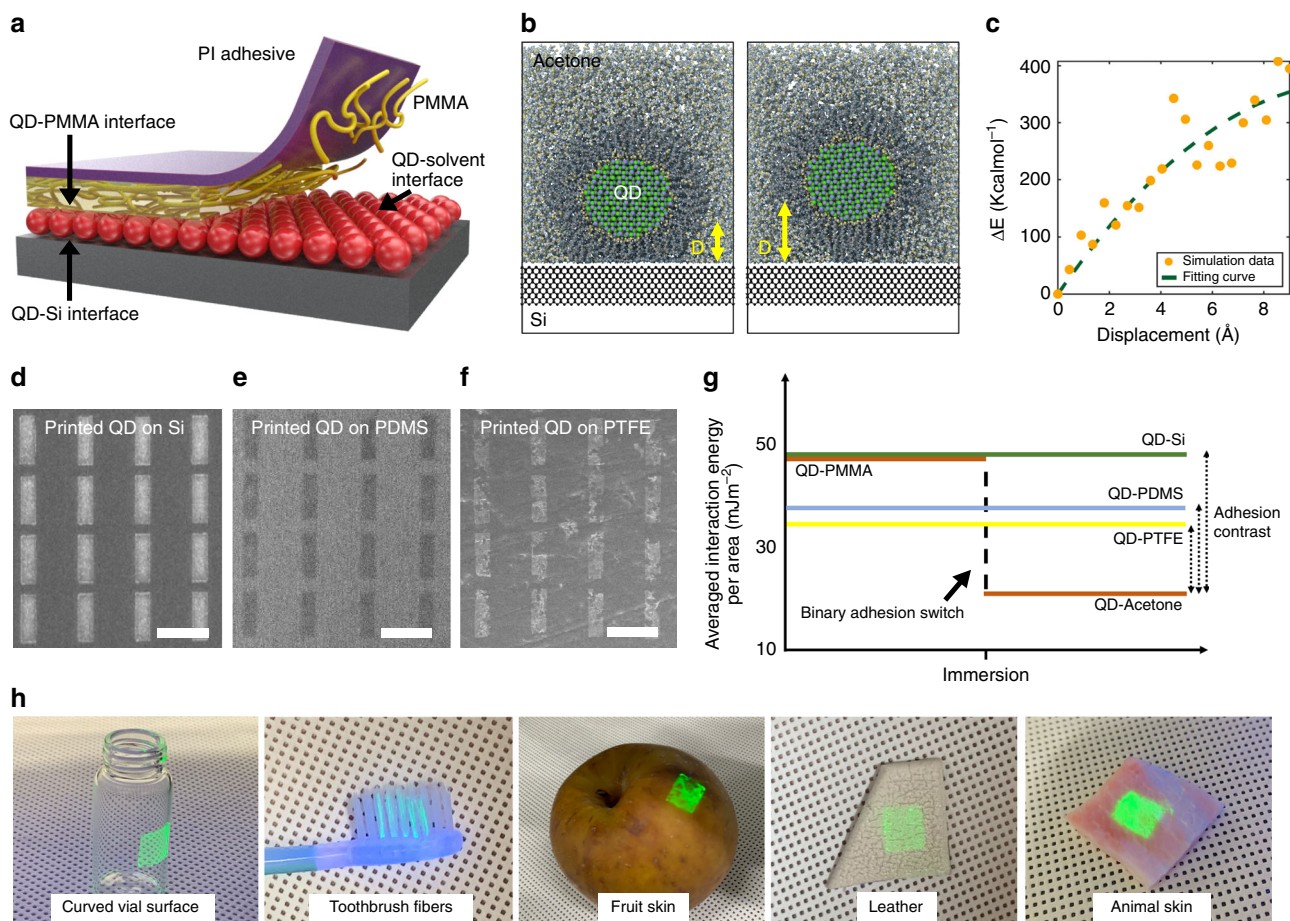

**Fig. 4 Non-kinetic iTP mechanism. a** Illustration of adhesion switch after PMMA dissolution in iTP mechanism. **b** MD simulation results of contacted (left) and lifted (right) QDs from the Si surface with a displacement D, when surrounded by acetone. **c** The simulation data of the excess energy necessary to lift the QD from the Si surface by the displacement D, is plotted and fitted. **d**–**f** SEM images of transferred QD arrays using pressure-free iTP on Si surface (**d**), PDMS surface (**e**), and PTFE surface (**f**). The printing demonstration on PDMS and PTFE surface indicate reversed predominance of adhesion via binary adhesion switching principle. Scale bars denote 10 μm. **g** The averaged interaction energy per area of the QD-media interfaces are calculated using MD simulation to illustrate the hypothetical relationship of adhesion in each interface as the PMMA dissolves during immersion. **h** QD pattern films printed on various surfaces with different curvature, topography, and hydrophobicity.

resolution of our QD pixel array shows 368 PPD, taking into account the 2-inch viewing distance of typical VR devices. A hexagonal pentile QD pixel array is also demonstrated to show versatility in pixel shape and arrangement (Fig. 5e). The RGB line profile of the polychromatic array (yellow solid line in Fig. 5b) is plotted in Fig. 5f, presenting good regularity of the emission of the printed QD patterns. As shown in Fig. 5g, RGB QD pixel arrays were delivered on a flexible PET substrate to show the adaptability for iTP for wearable/stretchable device applications.

Finally, an ELQLED device was fabricated by depositing a monochromatic QD film using iTP. We adopted the conventional indium tin oxide (ITO)/poly(3,4- ethylenedioxythiophene):polystyrenesulfonate (PEDOT:PSS)/poly[(9,9-dioctylfluorenyl-2,7-diyl)-co-(4,40-(N-(4-s-butylphenyl)) diphenylamine] (TFB)/QD/ZnO/Al device structure[49,50]. For fair comparison and investigation, three separate devices fabricated with the same device structure and QDs but composed of spun-cast, contact-printed, and iTP QD layers were prepared. The performances of reference spun-cast and contact-printed ELQLEDs fabricated by our group are consistent (slightly higher for contact-printed) with those of previous studies[18,20,50]. The left image of Fig. 5h shows the energy band diagram of a monochromatic green ELQLED device and the right image is taken during the operation. In

previous reports on printing based ELQLEDs, the device performance remained at an unsatisfactory level or was not fully supported by data. Figure 5i–k and Supplementary Fig. 13 demonstrate that our iTP ELQLED exhibits a maximum current efficiency of 14.8 cdA$^{-1}$ and a maximum EQE of 3.3%, which are comparable to those of spun-cast devices (high device performance but incapable of patterning) and outperform those of contact-printed devices. Supplementary Table 3 summarises the resolution and device performance data of previous studies reporting QD printing. To the best of our knowledge, the device performance reported herein is the highest among ELQLEDs fabricated using contact-printing reported to date and is superior to the best record by a factor of 37 (current efficiency) and 6.6 (EQE)[20,23,25]. This performance outcome suggests that the iTP minimised the degradation of the emission characteristics of QDs. Previous studies attributed the ELQLED performance degradation to charge quenching at interface defects[51–53]. Figure 5i shows that the iTP device exhibits higher luminance than that of a contact-printed device even at lower input current density. The excessive current density shown in contact-printed devices implies non-uniform film thickness and charge quenching defect points created by pressure-induced cracks of QD films. Other EL characteristics of the three devices such as turn-on-voltage, peak

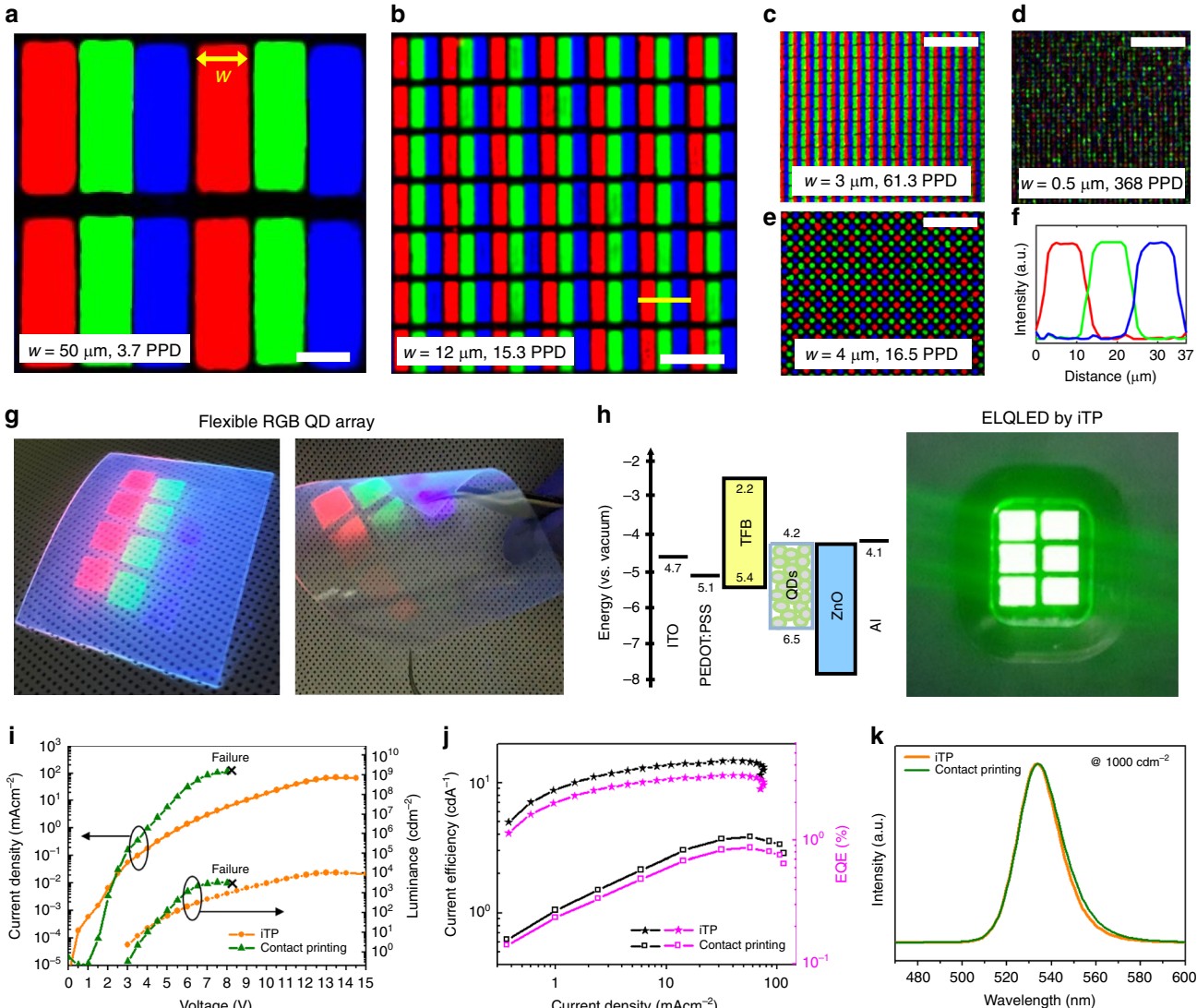

**Fig. 5 Fabrication of RGB full-color QD array and ELQLED device using iTP. a–e** Confocal fluorescence images of full-color RGB QD arrays in wide resolution range with highest resolution of 368 PPD (**d**). Feature width of subpixels, w, and resolution in PPD (viewing distance of ×1.5 the diagonal length of the display) are denoted in the images. Hexagonal pentile array design is also demonstrated (**e**). Scale bars denote 50 μm. **f** RGB profile is plotted across a single pixel of the QD array in **b** (profile marked in yellow) using imageJ. **g** RGB QD film arrays printed on flexible PET substrate using iTP. (**h**) Energy band diagram of the monochromatic green ELQLED fabricated using iTP (left), and photograph image of the operating device (right). **i** Current density–voltage–luminance (*J–V–L*) characteristics (symbol X indicates early failure of the contact-printed device), **j** Current efficiency and external quantum efficiency (*CE-EQE*) characteristics, and **k** Normalised electroluminescence (EL) spectra of ELQLED devices fabricated using iTP and contact printing.

wavelength, maximum luminance, and colour coordinates are summarized in Supplementary Fig. 14. Each characteristic indicates comparable values between the spun-cast and the iTP devices whereas clear shifts in the values and wavelength are observed in the contact-printed device. The high performance of our devices indicates that the pattern delivery process of iTP can securely preserve the characteristics of CTLs and their interfacial coherence between the constituent layers. At the current stage, copious studies have reported significant enhancement in the efficiency of ELQLEDs; however, it is critical to maintain the device performance even after the patterning process in order to realise full-colour ELQLED displays in practice. We believe the iTP technology could be a breakthrough to approach device performance maximization along with patterning capability with the goal of developing ultra-high resolution ELQLED displays.

## Discussion

This work demonstrates the highest-resolution full-colour RGB QD pixel array reported to date with outstanding pixel-density of up to 368 PPD. The newly developed iTP strategy enables omni-resolution printing capability even down to the ultimate resolution of the single-QD range. The printing yield of the iTP-fabricated QD array is ~100% for a wide resolution range. Deposition, patterning and delivering tasks of colloidal-state QDs are controlled via systematic engineering of binary solvent environments to realise a thermodynamic-driven, pressure-free printing mechanism. From the fact that the iTP is a printing-based technology, it is expected to be well suited with current roll-to-roll manufacturing that can even cover 3000 mm size substrates[54]. Also, to control the position of QD pixels, we used conventional mask-aligners, which are widely employed to

produce color-filter patterns in current display production lines. The iTP ELQLED records a maximum current efficiency of 14.8 cdA$^{-1}$ and a maximum EQE of 3.3% which are superior to those of previously reported devices based on contact-printing. We believe this new QD patterning technology establishes an effective and practical patterning tool for high-performance but laborious to handle colloidal QD inks while hardly compensating the degradation in device performance. Moreover, we expect that this iTP technique will be applied more generally to diverse other applications based on QDs such as biosensors, photovoltaic devices, catalysts, and so on.

## Methods

**QD synthesis**. *Preparation of CdSe/CdS/ZnS Red QDs*[55]: After 1 mmol of cadmium oxide (CdO), 4 mmol of oleic acid (OA), and 20 ml of 1-octadecene (ODE) were added into a 100 mL flask under an Ar condition, the temperature of the reactor was increased up to 300 °C. When the mixture became transparent, the 1st stock solution, which consists of selenium powder and trioctylphosphine (Se-TOP), was injected into the reactor for the growth of the CdSe core. The temperature of the reactor was then maintained at 300 °C for 90 s and 0.75 mmol of 1-dodecanethiol (DDT) was instilled in the reactor to cover the CdS shell on the cores. After 30 mins, a 2nd stock solution consisting of 4 mmol of zinc acetate (Zn(Ac)$_2$) and 2 mL of tributylphosphine (TBP) dissolving 4 mmol of sulfur powder (S) was added and reacted at the same temperature for 20 min.

*Preparation of CdSe/ZnS Green QDs*[56]: After degassing with vigorous stirring of 3.14 mmol of zinc oxide (ZnO), 0.14 mmol of cadmium acetate (Cd(Ac)2) and 7 mL of the OA were added to the 100 mL flask at 150 °C for 30 min. Then, 15 mL of ODE was added into the reactor. The reactor was changed to an argon condition and heated to 310 °C. To grow CdSe/ZnS QDs, 2.5 mmol of S, 1.5 mmol of Se and 3 mL of trioctylphosphine (TOP) were injected and held for 10 min. To form a ZnS shell, 1.6 mmol of S in 2.4 mL of ODE was added to the reactor. The temperature in the reactor was lowered to 270 °C after 12 min. At this temperature, a solution consisting of 3 mmol of zinc acetate dihydrate (Zn(Ac)2), 1 mL of OA and 4 mL of ODE was rapidly injected. In addition, the reactor was maintained for 20 min with the injection of a mixture of 9 mmol of S and 5 mL of TOP.

Preparation of CdZnS/ZnS Blue QDs[57]: 5 mmol of Zn(OA)2, 0.5 mmol of CdO, 3.5 mL of OA, and 7.5 mL of ODE were added in a 100 mL flask. To remove oxygen gas, degassing was carried out under a vacuum at 150 °C for 1 hr. Ar gas was then flowed and the temperature of the reactor was elevated to 310 °C. To nucleate the CdZnS cores, a 1st stock solution prepared by dissolving 0.9 mmol of S in the 1.5 mL of ODE was swiftly injected into the reactor. After 8 min, a 2nd stock solution prepared by mixing 4 mmol of S and 1.5 mL of TOP was slowly added into the reactor to grow a ZnS shell on the surface of the cores.

**Immersion transfer printing**. Pre-patterned Si templates were fabricated in various shapes and sizes. For line, rectangular, circular and diamond shape trenches, an 8-inch Si wafer was patterned using photolithography equipment (ASML, KrF Scanner (PAS 5500/700D) and then plasma-etched until a trench depth of 50 nm (LAM, TCP-9400DFM, ICP Type Si Etcher, Cl$_2$+HBr). A sub-10 nm SiO$_2$ template was fabricated by plasma etching self-assembled block copolymer patterns derived from poly(styrene-*b*-dimethylsiloxane) (PS-*b*-PDMS, Polymer Source Inc., MW = 36 kgmol$^{-1}$, $f_{PDMS}$ = 38%, SNTEK, ICP-RIE, O$_2$+CF$_4$). The surface of the pre-patterned templates was modified by spin-coating a short-chain hydroxyl-terminated poly(dimethylsiloxane) homopolymer (PDMS-OH, Polymer Source Inc., MW = 5 kgmol$^{-1}$, 2 wt% in heptane) onto Si hard templates under 3000 rpm for 20 s. The templates were then thermally treated in an oven under a vacuum for 30 min. The templates were sufficiently washed using heptane to remove excess PDMS-OH, resulting in 3–4 nm thick PDMS on the Si template. A colloidal QD solution was prepared by redispersing pre-synthesised red, green, and blue QDs in a heptane/toluene binary solvent of various compositions. The capillary force self-assembly of QDs was carried out by spin-coating the QD solution on prepared Si templates at a fixed rpm of 1500 for 10 s. For a sacrificial transfer medium, PMMA homopolymer was spin coated on the patterned QD substrate (PMMA, Polymer Source Inc., MW = 100 kgmol$^{-1}$, 2 wt% in acetone) at 3000 rpm for 10 s. A polyimide (PI) adhesive film was laminated on top of the PMMA and was peeled off to separate the QD patterns from the Si template. The Si template was reused for subsequent QD patterning. Lifted-off QD patterns were contacted on a bare Si substrate with minimal contact pressure of 2 kPa. For confocal microscope characterization, QD patterns were printed on a bare glass substrate. The substrate was then immersed in acetone (sufficient volume for complete immersion) for 35 min until the PI layer spontaneously separated from the target substrate. The target substrate was finally rinsed using acetone after being removed from the acetone bath.

**QD pixel alignment**. After a single colour QD array was printed on a Si substrate, the iTP process of second and third QD arrays emitting different colours was repeated as explained above. In order to align the subsequent arrays, the picked

QD/PMMA/PI layer was attached on glass substrate with PI layer at the bottom which was then placed on the mask holder of a mask aligner (Midas MDA-600S). The Si substrate with the first QD array was placed on the stage of the mask aligner and the arrays were manually aligned through x,y, tilt axis control. The aligned arrays were contacted under set pressure and the sample was removed from the aligner as contacted. The following immersion printing process was repeated, as explained above.

**QD pixel array characterization**. A scanning electron microscope (SEM, Hitachi S-4800) was used to obtain images of pre-patterned Si templates, self-assembled QD films in the templates, and printed QD pixel arrays on the target Si substrate. An atomic force microscope (Park Systems, XE-100) was used to characterise the thickness profile and morphology of the picked QD film on a PMMA surface and printed QD film on the target Si substrate. A confocal fluorescence microscope (Carl Zeiss LSM780) was used to obtain PL images of printed full-colour QD arrays.

**ELQLED fabrication and characterization**. The device structure of the fabricated QLED was indium-tin-oxide (ITO)/poly(3,4-ethylenedioxythiophene):poly(4-styrenesulfonate) (PEDOT:PSS, Clevios P VP AI4083)/Poly[(9,9-dioctyl-fluorenyl-2,7-diyl)-co-(4,4′-(N-(4-sec-butylphenyl)diphenylamine)] (TFB)/ CdSe/ZnS QDs (Green)/ZnO/Al. The ITO (120 nm) patterned glass substrate was cleaned using acetone, ethanol and isopropanol and dried at 120 °C for 30 min. After oxygen plasma treatment, PEDOT:PSS was spin-coated at 3000 rpm for 30 s and annealed at 120 °C for 30 min. Subsequently, 8 mgml$^{-1}$ of TFB solution in p-xylene was spin-coated at 3000 rpm for 60 s and annealed at 180 °C for 30 min. CdSe/ZnS QDs (green emission) were transfer printed onto the TFB film. For the reference device, the QDs were spin coated to achieve the same film thickness. ZnO in 2-methoxyethanol solution with 30 mgml$^{-1}$ was spin-coated at 2000 rpm for 60 s. All spin-coating and annealing processes were performed in a nitrogen atmosphere. Finally, an Al cathode layer was deposited to 120 nm thickness via a thermal evaporation method at a base pressure of $1 \times 10^{-7}$ Torr. The QLED device characteristics ($J$–$V$–$L$ and current efficiencies) were measured by a Keithley 2635 A source meter unit integrated with a Minolta CS2000 spectrophotometer.

**Molecular dynamics simulations**. The MD simulations were carried out using the Large-scale Atomic/Molecular Massively Parallel Simulator (LAMMPS)[58]. The PCFF+ force field was adopted to describe the interatomic interactions[59], with its parameters provided by the commercial simulation software MedeA (Si, PMMA, PDMS, Teflon and acetone) and the literature (Cd and Se)[43]. The PPPM method was used for solving long range Coulombic interactions, with modifications to properly treat the non-periodic condition along the $z$ direction[44]. Three types of configurations were generated separately first: a slab of (001) single crystalline Si was prepared for representing the Si substrate; to create each polymer substrate (PMMA, PDMS and Teflon), 150 chains with a chain length of 10 repeat units were randomly distributed and equilibrated at 300 K; a CdSe QD with its diameter of 4 nm was created, covered by 560 oleic acid ligands. These geometries were then merged to obtain the initial configurations of the MD simulations such that the passivated quantum dot was placed on top of the substrate. Under the NVT ensemble, simulations were performed with a time step of 0.5 fs for a time span of 5 ns. The interaction energy between the QD and the substrate was calculated by averaging the data from the last 500,000 steps. (To make a fair comparison, the substrate thickness for the calculations was kept the same at 1.2 nm for all the cases.) To estimate the energy change induced by lifting the QD from the surface, simulations were performed iteratively with constraining the center of mass position of the QD at different heights. Specifically, in each iteration, the QD was first dragged to move for a distance of 2 angstroms in 20 ps, and was allowed to relax for 140 ps. During these simulations, the bottom of the film and the internal coordinates of CdSe core were fixed.

## Data availability

The data that support the findings of this study are available from the corresponding author upon reasonable request.

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

## Acknowledgements

This research was supported by Creative Materials Discovery Program (NRF-2016M3D1A1900035) and the Global Frontier Program through the Global Frontier Hybrid Interface Materials (GFHIM) (2013M3A6B1078874) of the National Research Foundation of Korea (NRF) funded by the Ministry of Science and ICT.

## Author contributions

T.W.N. and Y.S.J. conceived the project and designed the experiments. T.W.N. conducted most of the fabrication and the analysis of experiments. M.K. conducted fabrication of ELQLED devices and the analysis of the devices. Y.W. and J.C.G. discussed and conducted

MD simulations. W.C. and D.Y.J. conducted QD syntheses. H.L. and G.Y.K. contributed to the PL analysis. K.M.S. contributed to the contact-angle analysis. T.W.N. and M.K. and Y.S.J. wrote most of the paper. All authors discussed the results and contributed to the final paper.

## Competing interests

The authors declare no competing interests.
