## [Peer Review File · Nature Communications]

Editorial Note: Editorial Note: This manuscript has been previously reviewed at another journal that is not operating a transparent peer review scheme. This document only contains reviewer comments and rebuttal letters for versions considered at Nature Communications.

REVIEWERS' COMMENTS:

Reviewer #4 (Remarks to the Author):

The manuscript by Yeon Sik Jung et al., reports on a QD patterning technologies with sub-micron feature size, high fidelity, and high post-patterning device performance. The results are impressive and were achieved thanks to the development of what the authors call thermodynamic-driven immersion transfer-printing (iTP). The work is very well detailed and maybe this is the biggest problem, namely that the manuscript reads extremely technical in some sections. I wonder if would have been possible for the authors to move some of the very technical sections in the supplementary information. I will also suggest the authors to enrich the citation of literature with the following alternative approach, DOI: 10.1002/admt.201900054, which in substance can also give a further support to their approach.

Reviewer #5 (Remarks to the Author):

I think the authors have revised the manuscript appropriately and addressed all major issues. Therefore, the manuscript is publishable in the current form.

■ **Point-by-point responses to the reviewers' comments.**

Reviewer #4

[General remarks] *The manuscript by Yeon Sik Jung et al., reports on a QD patterning technologies with sub-micron feature size, high fidelity, and high post-patterning device performance. The results are impressive and were achieved thanks to the development of what the authors call thermodynamic-driven immersion transfer-printing (iTTP). The work is very well detailed and maybe this is the biggest problem, namely that the manuscript reads extremely technical in some sections. I wonder if would have been possible for the authors to move some of the very technical sections in the supplementary information. I will also suggest the authors to enrich the citation of literature with the following alternative approach, DOI: 10.1002/admt.201900054, which in substance can also give a further support to their approach.*

[Comment 1] *I wonder if would have been possible for the authors to move some of the very technical sections in the supplementary information.*

[Response 1] We appreciate the reviewer's positive evaluation of our manuscript. We understand the concern raised by the reviewer that some of the sections in the manuscript are extremely technical. As a response, we modified the sections related to Figure 2 and 3 by rearranging partial information to supplementary information.

[Modification in the manuscript]

Figure 2 | Domain specific patterning of QD films by controlled dewetting mechanism. QD concentration and solvent composition dependent stability diagram of dried QD film are plotted separately for mesa (a) and trench (b) domains. The symbols ●, Δ, and □ indicate conformal QD film, heterogeneous dewetting near the mesa edge, and spinodal dewetting throughout the domains, respectively. (c) The domain coverage of QD film on mesa and trench area as a function of change in solvent composition. The blue shadow indicates experimental window of solvent composition in which the contrast in domain coverage between mesa and trench area is the greatest. (d-g) Tilted SEM images (respective schematics in the insets) that show the change in dewetting behavior of dried QD films as QD concentration decreases (constant toluene composition at 13%). Scale bars denote 300 nm.

Supplementary Figure 5 | Solvent stability analysis via effective interface potential calculation. (a) The calculated effective interface potential of heptane, toluene, and chloroform liquid film in Si/SiO_x/PDMS/solvent/air regime is plotted as a function of the solvent film thickness. (b) The second derivatives of effective interface potential of chloroform and toluene are plotted as a function of the solvent film thickness. The critical points marked as O and X indicate the critical film thickness, h_{crit} , under which spinodal dewetting occurs. (c) The calculated characteristic wavelength λ_s of toluene is plotted as a function of the solvent film thickness. AFM image (left) and corresponding FFT image (right) of dry QD film spun-cast under binary solvent composition regime are shown in the inset.

“The effective interface potential curve shown in Supplementary Figure 5(a) suggests ...” On page 8.

“According to Supplementary Figure 5(b), h_{crit} for toluene and chloroform was calculated to be 5.63 nm and 3.95 nm, respectively.” On page 9.

“As shown in Supplementary Figure 2(c), we also confirmed consistency between ...” On page 9.

[Comment 2] *I will also suggest the authors to enrich the citation of literature with the following alternative approach, DOI: 10.1002/admt.201900054, which in substance can also give a further support to their approach.*

[Response 2] We thank the reviewer’s recommendation on an additional literature which can broaden the scope of the related research in the field of QD patterning. We provided the additional citation in the introduction section of our manuscript, which we hope provides greater insight to the readers of the journal.

[Modification in the manuscript]

“Several alternative fabrication techniques, such as transfer-printing, inkjet printing, and lithography^{17,19-30} have been utilised to demonstrate the patterning of colloidal QDs with the goal of improving the performance of conventional displays.” On page 3.

Additional citation

[24] Shulga, A. G. *et al.* Patterned Quantum Dot Photosensitive FETs for Medium Frequency Optoelectronics. *Adv. Mater. Technol.* **4**, 1900054, doi:10.1002/admt.201900054 (2019).

Reviewer #5

[General remarks] *I think the authors have revised the manuscript appropriately and addressed all major issues. Therefore, the manuscript is publishable in the current form.*

[Response] We appreciate the reviewer’s positive evaluation for the previously revised manuscript. We hope the final version of our manuscript successfully delivers the new innovation of our work to the readers of the journal.